# Design and Implementation of Airway Response Teams to Improve the Practice of Emergency Airway Management

**DOI:** 10.3390/jcm11216336

**Published:** 2022-10-27

**Authors:** Kelly A. Tankard, Milad Sharifpour, Marvin G. Chang, Edward A. Bittner

**Affiliations:** 1Department of Anesthesia, Critical Care and Pain Medicine, Massachusetts General Hospital, Boston, MA 02114, USA; 2Department of Cardiac Surgery, Cedars-Sinai Medical Center, Smidt Heart Institute, Los Angeles, CA 90048, USA

**Keywords:** airway response team, emergency airway management, intubation-related complications, difficult airway

## Abstract

Emergency airway management (EAM) is a commonly performed procedure in the critical care setting. Despite clinical advances that help practitioners identify patients at risk for having a difficult airway, improved airway management tools, and algorithms that guide clinical decision-making, the practice of EAM is associated with significant morbidity and mortality. Evidence suggests that a dedicated airway response team (ART) can help mitigate the risks associated with EAM and provide a framework for airway management in acute settings. We review the risks and challenges related to EAM and describe strategies to improve patient care and outcomes via implementation of an ART.

## 1. Introduction

Emergency airway management (EAM) is commonly performed in the hospital setting. However, despite advances in identifying patients at increased risk for having a difficult airway, improved airway management tools, and algorithms to assist with EAM decision-making, there remains significant morbidity and mortality associated with EAM [1,2]. Evidence suggest that a dedicated airway response team (ART) can reduce the risks associated with EAM and improve patient outcomes [3,4,5,6]. Here, we review the risks and challenges related to EAM and describe strategies and key elements to improve patient care via implementation of an ART (Figure 1).

## 2. Risks Associated with EAM

Risks associated with EAM include physiological complications (e.g., hypoxemia and hemodynamic instability), traumatic complications (e.g., dental and soft tissue injury) and procedure-related complications (e.g., esophageal intubation and aspiration) (Table 1) [9,10]. The complication rate of EAM outside of the operating room (OR) setting exceeds 45% [1]. Hypotension and hypoxemia occur frequently during EAM and have been associated with increased mortality even after adjusting for other variables [1,11,12]. In 2011, the 4th National Audit Project of the Royal College of Anaesthetists (NAP4) published a comprehensive examination of major complications associated with airway management in the United Kingdom which found 40- and 50-fold higher risks of hypoxic brain injury and mortality for airway management performed in the emergency room and intensive care unit (ICU), respectively, when compared to routine intubation in the OR [2]. Subsequent analysis by the NAP4 investigators identified provider inexperience, lack of adherence to EAM guidelines, and absence of a clear contingency plan in the event of failed initial attempts as factors responsible for 40% of the adverse outcomes [9]. While a follow-up survey showed that many UK hospitals had instituted changes based on the NAP4 findings, a “safety gap” remains between ideal practice and what occurs in actual practice [13]. A recent analysis of EAM related complications included in the American Society of Anesthesiologists (ASA) closed-claims database also showed that human factors played a prominent role in EAM-related complications [14]. The closed claims analysis concluded that the majority of death or permanent brain damage related to difficult endotracheal intubation occurred due to insufficient knowledge (not recognizing risk factors for difficult airway management or not knowing the guidelines), systems failures (rescue equipment or personnel not being available), and delays in decision-making (such as progression to cricothyrotomy), and were preventable [15]. Based on these findings, strategies to reduce the risks of EAM are urgently needed.

## 3. Challenges in EAM

The risks associated with EAM are multifactorial and are attributed to patient-related, procedure-related, and environmental-related factors (Figure 1) [7]. Patients who require EAM are often hemodynamically unstable and may not tolerate the hemodynamic perturbations associated with endotracheal intubation. Efforts to provide preoxygenation prior to intubation may be difficult or less effective due to patient agitation, hemodynamic instability, and ventilation-perfusion mismatch. Less effective preoxygenation limits the amount of time available to secure the airway before the development of dangerously low oxygen saturation levels and cardiopulmonary collapse. Poor tissue perfusion may result in inaccurate or unmeasurable oxygen saturation (SpO2) readings, which further confound patient assessment and monitoring. Hypoxemia, hypercarbia, and respiratory distress result in increased sympathetic outflow, which elevates blood pressure and may result in a false sense of “hemodynamic stability”, followed by an exaggerated drop in blood pressure once the airway is secured. Induction medications, given to facilitate intubation, can worsen hypotension by decreasing systemic vascular resistance and depressing myocardial contractility. Patients requiring EAM often have full stomachs and are at increased risk for aspiration of gastric contents. Anatomical abnormalities such as cervical spine instability, orotracheal or laryngeal tumors, history of neck radiation, airway edema or stenosis, or prior airway instrumentation are additional risk factors for adverse events during EAM. Lastly, the increasing prevalence of obesity and the number of patients who receive anticoagulation increase the risk of EAM-related complications.

Although difficult or failed orotracheal intubation is typically the focus of EAM strategies, acute airway dislodgement, bleeding, or obstruction of an existing surgical airway (e.g., tracheostomy or laryngectomy) can pose equally daunting challenges [16]. Critical airway events frequently occur in patients with previously secured airways. In one series, 82% of the critical airway events occurred after intubation, with 25% of these events contributing to patient deaths [17].

The clinical environments in which EAM is performed pose yet another challenge; airway emergencies can occur in remote places, resulting in delayed response times. Furthermore, the responding clinicians may not know the bedside providers and the resources that are available to them, which may result in suboptimal communication and division of tasks. Patient positioning is more difficult on a hospital bed or in a CT scanner compared to an OR table, resulting in limited access to the patient and suboptimal positioning, which make airway management more difficult. Similarly, EAM often occurs after hours, when fewer clinicians are available to provide assistance [18]. In addition, clinicians responding to EAM are often less experienced than providers who perform routine airway management in the OR [1,19,20]. Given the instability of many patients requiring EAM, delaying management until more experienced staff are available is often not feasible. The risks, challenges, and complications associated with EAM justify the creation of dedicated ARTs.

## 4. Rationale for an Airway Response Team

The importance of an organized, systematic, and team-based approach to patient care in emergency settings is well-established and serves as the basis for programs like Advanced Cardiovascular Life Support (ACLS) [21]. Nonetheless, EAM has historically been managed by a single clinician, often a trainee, rather than a team [22]. The desire to improve patient safety and mitigate the risks and complications associated with EAM, as described above, has led to creation of ARTs in many institutions [3,4,5,6,23,24,25,26,27]. There is growing evidence that the team approach to EAM can improve outcomes by reducing patient complications, death, and malpractice claims [3,4,5,6,25].

The ART model highlights the different skills that are required for successful EAM. As such, ARTs are often multidisciplinary teams that include respiratory therapists, nurses, anesthesiologists, and surgeons, and may include both trainees and attending physicians [28]. Highly effective ARTs should have the ability to immediately convert to a surgical airway without delay in the event of a failed intubation. Since anesthesiologists do not routinely perform surgical airways and many graduating surgical trainees have never performed an emergency surgical airway [15,29], a surgeon trained in performing surgical airways should be available in the event of a failed mask ventilation and intubation [27].

## 5. The Team Model

The main objective of the ART is to rapidly assemble team members and resources to assess and manage a patient with an airway emergency. Having a designated team of clinicians with airway expertise and a method for activation that alerts and assembles the team for EAM in advance of an intervention is a distinctly superior approach to “calling for help” in an emergency after a practitioner has unsuccessfully attempted to manage an airway. The Difficult Airway Society has proposed several models for an ART, suggesting that the team ideally includes two clinicians with advanced airway management skills (intubation) [30]. The delegation of tasks and roles depends on how many members are on the ART. For a smaller group, one team member may need to take on multiple roles. Additionally, depending on the number and clinical experience of the team members, the duties may change during EAM. For example, if there is a failed first attempt at intubation, the second (ideally more experienced) clinician may take over. There should also be a designated team leader who guides the team. As previously stated, a clinician (typically a surgeon) who is trained in emergency front of the neck access (FONA) should be readily available. However, providing 24/7 in-house coverage with a surgeon who is trained in FONA often poses a challenge to the ART model, especially in smaller hospitals with fewer resources. Depending on the institutional resources, there are a range of clinical specialists, including emergency medicine physicians, intensivists, and anesthesiologists who may be trained to be proficient in surgical airway techniques if 24/7 in-house surgical coverage is not possible. Nurses, pharmacists, respiratory therapists, and a technician capable of providing and managing the equipment required for EAM are the other vital components of the team (Figure 2). The patient’s primary nurse and medical team should also be considered part of the ART, as they can provide important contextual information relating to the patient’s history and clinical course and can communicate with and provide comfort to the patient during EAM. Lastly, hospital administrators are also considered part of the ART, and their role is to invest in resource allocation, training programs, and education of ART staff. Telemedicine may play an emerging role in managing ARTs in settings where expertise is not readily available [31,32].

## 6. Activation

The ART is valuable not only for proceeding with emergent airway interventions but also for assessment, triaging, and planning of patients with airway issues [28]. While an ART often provides immediate airway intervention either by intubation or surgical airway, other valuable roles of the ART include triaging a patient to the OR for airway management, assessment and determination that no immediate action is necessary, formulating a plan should an airway management be required in the future, or assessing a patient with a history of difficult intubation for readiness prior to extubation (Figure 3).

Timely and appropriate activation of the ART are essential to ensure the best patient outcomes. Providing clinicians with clear criteria for ART activation and a consistent way to notify the team is key to success. Strategies to educate hospital staff on how and when to activate the ART include in training sessions, information sheets posted in key locations throughout email notifications, and required web-based training. Implementing an ART will require educating care teams as well as the ART responders. Staff should be educated on what information to provide to the ART at the time of activation (Figure 4). A system should be developed to monitor the timeliness and appropriateness of ART notifications and feedback should be provided to bedside providers and team members to improve performance.

## 7. Identification of High-Risk Patients

Establishing airway screening practices and implementing policies to care for patients with suspected or documented difficult airway are important systems-level strategies for preventing EAM-associated complications. Since many patients with predictably difficult airways go undiagnosed, bedside clinicians should be educated on performing a basic airway assessment. Patients who are identified to have risk factors for a difficult airway may benefit from an ART consultation prior to undergoing sedation, non-emergent intubation, or extubation in a non-OR setting, which may potentially prevent airway crises. Documentation of risk for difficult airway in the electronic medical record provides readily accessible information as another safeguard to improve patient safety.

While airway examination is critical to identify patients at risk for difficult intubation, it is often not performed in patients undergoing EAM [30]. Although comprehensive airway examination may not be feasible, a basic airway examination can typically be performed. The MACOCHA score is a validated tool for early identification of patients at increased risk of difficult intubation in the intensive care unit [34]. It consists of patient related risk factors (Mallampati score of 3 or 4, obstructive sleep apnea (OSA), reduced cervical spine mobility, limited mouth opening), operator related risk factors (non-anesthesiologist intubating clinicians), and indicators of disease severity (coma, severe hypoxia) (Table 2). Five points are assigned for the Mallampati score, two points for OSA, and one point each for the remaining components; a MACOCHA score less than three is predictive of a lower risk of difficult intubation. Ultrasound is becoming an increasingly popular tool for surgical airway planning as it can assist with anatomical identification of the cricothyroid membrane, vascular structures, and thyroid and may have a role in EAM if time permits [35].

While communication of information regarding difficult airway is important for preventing airway emergencies, consistent and reliable communication of such information does not always occur in the hospital setting. Several strategies have been proposed to improve communication of important airway information including placing difficult airway alert signs above a patient’s bed, airway alert bracelets worn on the wrist, stickers placed on the endotracheal tube of intubated patients to remind providers of difficult airway, and airway alert notifications incorporated into the medical record [33,36,37]. Utilizing multiple strategies simultaneously maximizes the likelihood that important airway information will not be missed. Handoff between providers about patients with airway risk should include information about airway management history, risk factors, anticipated difficulties, and contingency plans if airway management is needed.

Intubation is not the only critical time for patients with a known history of, or at risk for difficult airway; extubation is often overlooked by clinicians as a time of increased risk in patients that may be difficult to reintubate [38]. Plans for extubating high-risk patients should include consultation with the ART. In patients at the highest risk for extubation failure and difficult reintubation, a member of the ART with airway management experience should be present. If airway difficulty is anticipated, the presence of the ART at bedside, or a plan for extubating in the OR with preparations for a surgical airway may be warranted.

## 8. Equipment Availability

Lack of appropriate equipment and unfamiliarity with the existing equipment are risk factors that contribute to adverse events in EAM [2,39,40]. Equipment should be readily available, working properly, and familiar to ART clinicians. Ensuring airway equipment availability may be facilitated by using portable airway equipment bags or carts that store vital airway equipment [40,41,42]. Since different brands of commonly used airway devices can have notable differences, standardization of airway equipment is essential as the emergent setting is not the optimal time for staff to familiarize themselves with new devices. Standardizing locations and equipment eliminate unfamiliarity and allow for more effective ART utilization [40,41,42]

In addition to standard airway equipment, a kit of supplies for performing FONA should be immediately available. Point-of-care ultrasound is increasingly being utilized to facilitate landmark identification for surgical airway access. Therefore, a portable ultrasound device should be readily available [35]. Ensuring the consistent availability of airway equipment throughout an institution can be logistically challenging and reliable processes are needed for restocking supplies and keeping equipment clean and in working condition. Single-use products (laryngoscopes, bronchoscopes, and FONA kits) may be cost effective by eliminating some of the challenges of equipment maintenance.

## 9. Teamwork

Successful teamwork requires clear role designation, closed-loop communication, and ability to adapt to changes in the patient’s clinical condition and the environment [43,44,45]. The team leader should initiate an airway huddle when time permits, to introduce the time members, assign roles, discuss airway management strategy including primary and alternate plans, verify equipment availability and functionality, and provide an opportunity for team members to ask questions and voice concerns [43,44]. The team leader should be free of hands-on clinical responsibility to focus on team management and execution of the care plan. Delegation of tasks by the team leader prevents confusion. Because hierarchies can impair optimal team performance, all team members should be encouraged to speak up, voice concerns, and offer ideas via closed-loop communication [43].

Cognitive aids can provide a systematic framework to support the ART during EAM. These tools may include flowsheets, checklists, protocols, and visual aids to improve communication, remind team members of algorithms, and to ensure steps are not missed or overlooked [46,47,48,49]. These tools may be especially useful for clinicians with less experience [46]. They provide cognitive unloading, help to streamline decisions, ensure diagnoses and treatments are not missed, and offer team members the opportunity to speak up and request help.

Once the airway is secured, a summary of the ART’s decision-making process, interventions performed, anticipated airway problems should be provided verbally to the patient’s primary clinical team and documented in the medical record. In addition, information should be provided on how to contact the ART if questions or concerns arise [33].

## 10. Implementation

Implementing the ART in an institution requires policy development, educating clinicians, maintaining and supplying equipment, tracking utilization, and reviewing practice for quality improvement. A multidisciplinary airway safety committee composed of clinicians who are invested in improving airway management should be established, with an individual designated as the airway lead, who can serve to coordinate ART implementation, manage ART related clinical issues, and act as the liaison between the ART and hospital leadership [50]. Performing an initial gap analysis is necessary to identify areas where improvements to airway practices are most needed [13]. The early goals of the ART program should be realistic and achievable within a defined time period. These goals should be institution specific and depend on the existing level of airway management practices. The newly implemented ART should address airway management needs and not replicate the roles of other acute response teams. Working closely with other clinical emergency committees (e.g., code committee) will help to optimize team roles and practice within an institution.

The ART should be introduced to clinicians across specialties and locations throughout the hospital. Posters, email alerts, and web-based training sessions are methods that may be useful for disseminating information about the roles and activation methods of the ART. Simulation exercises can increase awareness, educate staff, and evaluate the functioning of the ART activation within clinical units [51,52]. Social media can also be utilized effectively to raise awareness of the ART roles and improve EAM education [53,54].

Anticipation of the costs of implementation and maintenance of an ART is essential for the program to remain financially viable. Key financial variables to consider include: (a) costs of airway equipment and medications; (b) the estimated volume of airway consults and procedures; (c) personnel costs and (d) reimbursement associated with team practice [55]. Equipment costs to account for include protective personal equipment (i.e., surgical masks, face shield, gown, gloves), disposable airway equipment (e.g., laryngoscope blades and endotracheal tubes, suction catheters, oral airways, SGAs and bag-mask assemblies), medications (i.e., sedatives, neuromuscular blocking agents, and intravenous vasoactive medications), as well as capital expenditures for new equipment such as airway carts with equipment for advanced airway management (i.e., fiberoptic bronchoscopes and video-laryngoscopes). Personnel costs will be based on ART composition as well as the extent to which team members may be engaged in other clinical roles. The volume of emergency airway calls prior to ART implementation will provide information on whether ART members should be solely dedicated to airway responsibilities or may be given additional assignments. ARTs may also serve as consultants for performing other procedures (e.g., obtaining arterial and central venous access) and reimbursement for these services will impact cost considerations. Ongoing education costs, communication equipment or administrative support for data collection and quality improvement should also be considered. The average reimbursement for an emergency airway management consultation multiplied by the frequency of emergency consults should provide a reasonable estimate of total ART reimbursement. Such financial analysis will require periodic reassessment after ART implementation. Sharing the financial costs of the ART program across multiple departments may increase buy-in. Hospital administration may challenge the need for this investment unless it can be adequately justified. Many ARTs have arisen from quality improvement initiatives, where an institutional airway safety problem was identified, and a team was established to address it. Monitoring ART activations, critical airway events, and malpractice claims over time may be the most effective method for demonstrating the value of the ART to the hospital administration, as a decreasing number of adverse events and claims demonstrate patient benefit while reducing costs [28].

## 11. Education

Education is an essential element of successful ART implementation. ARTs should provide education for airway management skill development, new airway management equipment, algorithms, and improving communication and teamwork [19,28]. Despite this, many institutions do not offer formal education regarding EAM or provide limited education to the trainees [22,56,57]. Furthermore, the EAM education provided within different clinical specialties often occurs in isolation and does not overlap. This may lead to differences in terminology, management approaches, and lack of team cohesion when different specialties converge to manage an airway emergency. To address this, educational efforts should focus on a multidisciplinary approach across specialties in order to share knowledge and break down barriers and hierarchies [58]. Evidence indicates that clinicians that train together perform better, are more adaptable, and improve patient outcomes [59,60].

The development of clinical expertise by an ART requires repeated practice in diverse and challenging situations. It is increasing recognized that skill development in EAM cannot be achieved through clinical practice alone. Simulation is playing an increasing role in ART training, demonstrating accelerated and sustained learning outcomes [61,62]. Simulation-based airway management training can improve knowledge and technical skills, and improve patient outcomes compared with traditional learning strategies [63]. Similarly, simulation-based training can improve the clinical acquisition and maintenance of infrequently employed but essential FONA skills [64,65,66]. In addition, simulation-based training is particularly effective for developing nontechnical skills required for effective team practice in crisis management, including leadership, decision making, and communication [67].

While high-fidelity simulators allow participants to immerse themselves in clinical scenarios that look and feel real, these simulators are resource intensive and often require that clinicians take time away from their clinical responsibilities to participate. Alternatively, in situ simulation, in which simulation is incorporated into the activities of daily practice, is less disruptive to staff scheduling and offers teams the opportunity to test their effectiveness in the actual clinical environment [52,68]. To maximize effectiveness, in situ simulation training should include ART members and other clinicians and utilize locally available equipment and resources. As team members become comfortable with this simulation training approach, they should be challenged with more challenging scenarios to provide a degree of stress inoculation reflective of the actual practice. In situ simulation has been demonstrated to improve process optimization, systems learning, and team dynamics [69].

Surgical airway training is an educational priority for ART practice. There are a variety of approaches to obtain FONA including both needle and scalpel-based techniques [70,71]. Training for FONA on cadavers has been used successfully to enhance realism but cost and availability may limit widespread use [72,73]. Another strategy for clinicians to gain experience with surgical airways is to participate in elective surgical airways in the OR [56]. Although FONA is not often required, having clinicians who are proficient in performing the procedure can be lifesaving when needed. The assumption that a single provider should be expected to be proficient in the full range of airway management skills, including the ability to establish FONA, deserves consideration in the context of an institution’s available and skilled personnel [56]. Training to obtain FONA skills, while essential, reduces the amount of time available for training in other advanced airway skills such as supraglottic airway placement (SGA; laryngeal mask airway, combi-tube), fiberoptic intubation, and video laryngoscopy, which, if used effectively, reduce the likelihood of needing FONA. In particular, ensuring skill with SGA placement is crucial as they are easy-to-use rescue tools for establishing and maintaining a patent airway and may be unfamiliar to non-Anesthesiology trained providers [74]. The specialized Combitube SGA, equipped with a distal cuff designed to prevent gastric inflation and aspiration, may be particularly suited to use in EAM [75].

## 12. Monitoring and Quality Improvement

ART activation and utilization should be regularly monitored to identify challenges to successful practice and opportunities for improvement [25,28]. Data should be collected on activation details (times, locations, ART response time), consulting teams, airway equipment utilized, equipment issues, communication challenges, patient outcomes, and successful activations and sentinel events. Documenting ART success is key to demonstrating the impact of the resources invested in the ART program on patient outcomes. Collected data from each ART activation should be compiled to facilitate longitudinal and in-depth analysis for subsequent optimization.

Safety concerns should be discussed in with the clinicians involved to identify systems-based issues that would benefit from intensive review or to initiate immediate process improvement to minimize the risk of recurrence in the future. Team members should be provided with support in case of an adverse patient outcome. Safety events involving the ART should also be reviewed by the airway committee to allow for more comprehensive multidisciplinary evaluation and to develop priorities for practice improvement.

## 13. Summary

Airway emergencies are common in the acute care setting and ineffective response is a source of preventable patient harm. Complications can be minimized by implementation of an ART program, which provides an organized, multidisciplinary, team-based approach to EAM. At a minimum, an effective ART includes a cohesive group of clinicians with airway expertise, including skills to perform FONA, and clear criteria for activation. By establishing clear criteria for activate an ART, educating clinicians to identify high-risk patients, ensuring equipment availability, optimizing teamwork, and providing education, the ART can improve care and create a safer environment for both patients and clinicians (Figure 5).

## Figures and Tables

**Figure 1 jcm-11-06336-f001:**
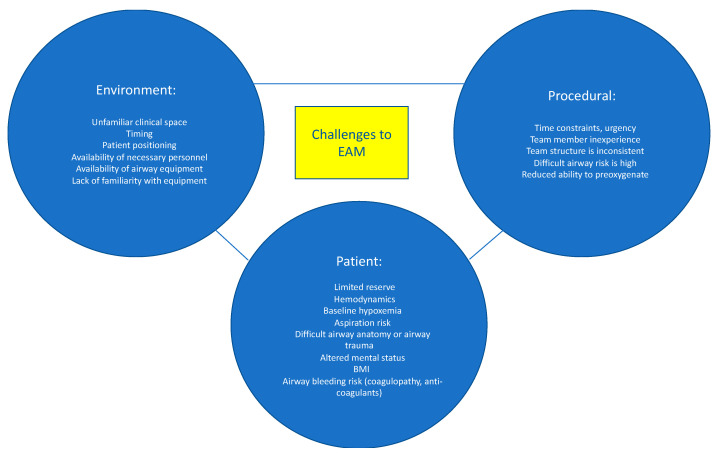
Challenges in Emergency Airway Management. Data from Higgs et al. [7] and Jabaley and Bittner [8].

**Figure 2 jcm-11-06336-f002:**
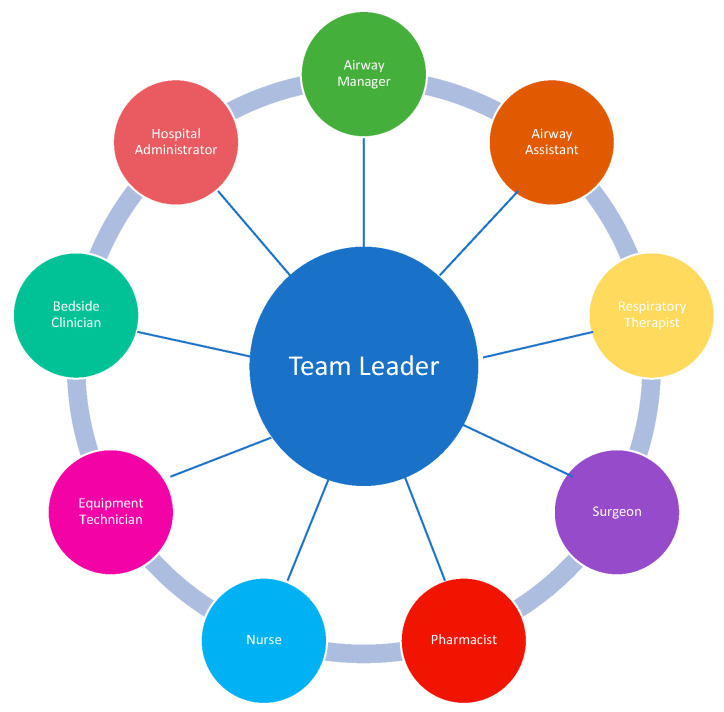
ART Team Members. Data from Long L, Vanderhoff B, Smyke N et al. [23] and Damrose JF, Eropkin W, Ng S et al. [24].

**Figure 3 jcm-11-06336-f003:**
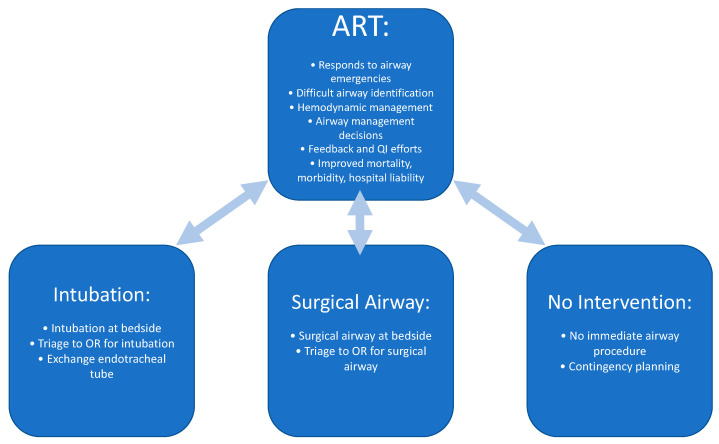
Roles of the Airway Response Team. Data from Atkins JH, Rassekh CH, Chalian AA et al. [25] and Atkins JH, Rassekh CH [27].

**Figure 4 jcm-11-06336-f004:**
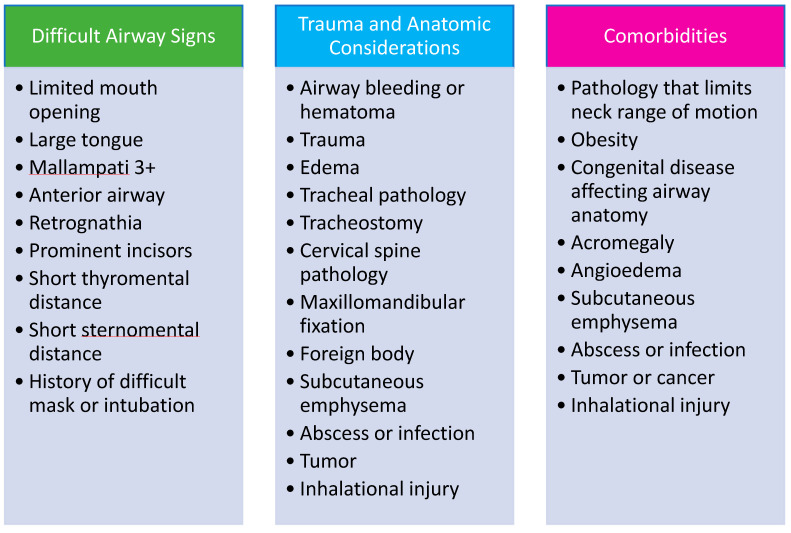
Characteristics of High-Risk Patients and Factors Suggestive of Difficult Airway Management. Data from Damrose JF, Eropkin W, Ng S et al. [24], Atkins JH, Rassekh CH, Chalian AA et al. [25], Atkins JH, Rassekh CH [27] and Feinleib J, Foley L, Mark L. [33].

**Figure 5 jcm-11-06336-f005:**
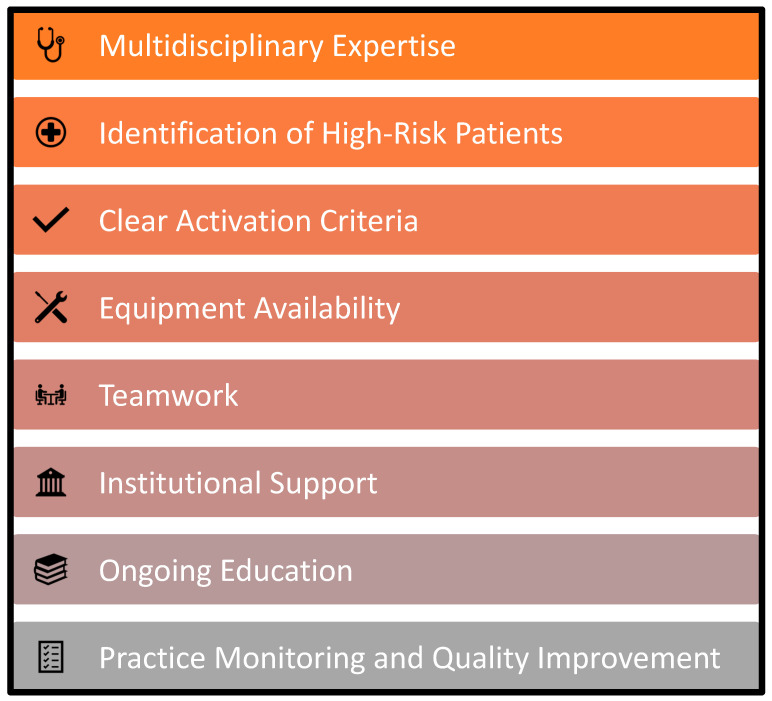
Key elements for effective airway response teams. Adapted from Bittner and Sharifpour [28].

**Table 1 jcm-11-06336-t001:** Complications of Emergency Airway Management.

Physiologic Complications	Traumatic Complications	Procedure-Related Complications
▪ Hypoxemia ▪ Hypercarbia ▪ Hypotension ▪ Hypertension ▪ Arrhythmias ▪ Cardiac Arrest	▪ Dental injury ▪ Soft tissue injury ▪ Arytenoid cartilage dislocation ▪ Vocal cord injury ▪ Temporomandibular joint dislocation ▪ Cervical spine injury ▪ Tracheal injury ▪ Barotrauma ▪ Bleeding	▪ Aspiration ▪ Gastric insufflation ▪ Bronchospasm ▪ Laryngospasm ▪ Esophageal Intubation ▪ Mainstem intubation

Adapted from Cook TM and MacDougall-Davis SR [9] and Hagberg C, Georgi R, Krier C [10].

**Table 2 jcm-11-06336-t002:** The MACOCCHA Score.

Factors	Points
Patient-related	
▪ Mallampati III or IV	5
▪ Obstructive sleep apnea	2
▪ Reduced cervical spine mobility	1
▪ Limited mouth opening (<3 cm)	1
Pathology-related	
▪ Coma	1
▪ Severe hypoxemia (<80%)	1
Operator-related	
▪ Nonanesthesiologist	1

Adapted from De Jong A et al. [34].

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
