# Peer review of "Design and Implementation of Airway Response Teams to Improve the Practice of Emergency Airway Management"

_jcm, 2022, doi:10.3390/jcm11216336_

Round 1

Reviewer 1 Report

This is an excellent and very important article comprising almost all aspects improving emergency airway management. I propose to include two more studies and describe them shortly in the manuscript:

Training for cricithyrotomies may also be done in cadavers wherever possible:

Eisenburger  et al , Comparison of conventional surgical versus Seldinger technique emergency cricothyrotomy performed by inexperienced clinicians. Anesthesiology 2000;92:687-90. doi: 10.1097/00000542-200003000-00012.

The Combitube should be mentioned as a lesser-known airway management device because it is easy to learn and is both safe against aspiration and efficient in ventilation:

Mort TC. Laryngeal Mask Airway and Bougie Intubation Failures: The Combitube as a Secondary Rescue Device for In-Hospital Emergency Airway Management. Anesth Analg 103:1264 –6, 2006.

Author Response

Reviewer 1:

This is an excellent and very important article comprising almost all aspects improving emergency airway management. I propose to include two more studies and describe them shortly in the manuscript:

Thank you for your kind comments and thoughtful review of our manuscript.

Training for cricithyrotomies may also be done in cadavers wherever possible:

Eisenburger  et al , Comparison of conventional surgical versus Seldinger technique emergency cricothyrotomy performed by inexperienced clinicians. Anesthesiology 2000;92:687-90. doi: 10.1097/00000542-200003000-00012.

Thank you for the suggestion. We have added the following text accordingly:

Training for FONA on cadavers has been used successfully to enhance realism but cost and availability may limit widespread use.  

The Combitube should be mentioned as a lesser-known airway management device because it is easy to learn and is both safe against aspiration and efficient in ventilation:

Mort TC. Laryngeal Mask Airway and Bougie Intubation Failures: The Combitube as a Secondary Rescue Device for In-Hospital Emergency Airway Management. Anesth Analg 103:1264 –6, 2006.

Thank you for the suggestion. We have added the following text accordingly:

In particular, ensuring skill with SGA placement is crucial as they are easy-to-use rescue tools for establishing and maintaining a patent airway and may be unfamiliar to non-Anesthesiology trained providers. The specialized Combitube SGA, equipped with a distal cuff designed to prevent gastric inflation and aspiration, may be particularly suited to use in EAM.

Reviewer 2 Report

Excellent paper.  Well thought out and organized.  Brings a lot of issues to the forefront in emergency airway management outside of the OR suites.  Can certainly be used as a primer for hospital systems looking to improve their emergency airway management teams.  Some thoughts:

Lines 33-37 likely need some references

LIne 41-define more clearly what are procedure-related complications.  This sounds like you are describing OR procedures.  Would be better if you defined more as bedside branch, urgent GI scope, etc.  

Lines 66-69 seems like a run on sentence.  I think the sentence structure could be improved.

lines 162-171. Having a table/figure showing the MACOCHA score would help make the paper stronger and clearer.  

I think a paragraph on associated costs (financial; personnel, etc). would make the the paper even stronger and give more food for thought.

Author Response

Reviewer 2:

Excellent paper.  Well thought out and organized.  Brings a lot of issues to the forefront in emergency airway management outside of the OR suites.  Can certainly be used as a primer for hospital systems looking to improve their emergency airway management teams.  Some thoughts:

Lines 33-37 likely need some references

Thank you for the suggestion. We have added additional references

LIne 41-define more clearly what are procedure-related complications.  This sounds like you are describing OR procedures.  Would be better if you defined more as bedside branch, urgent GI scope, etc.  

Thank you for the suggestion. We have added the additional text and Table 1 for clarification

Risks associated with EAM include physiological complications (e.g. hypoxemia and hemodynamic instability), traumatic complications (e.g. dental and soft tissue injury) and procedure-related complications (e.g. esophageal intubation and aspiration) (Table 1).

Lines 66-69 seems like a run on sentence.  I think the sentence structure could be improved.

Thank you for the suggestion. We have revised the sentence accordingly:

Efforts to provide preoxygenation prior to intubation may be difficult or less effective due to patient agitation, hemodynamic instability, and ventilation-perfusion mismatch. Less effective preoxygenation limits the amount of time available to secure the airway before the development of dangerously low oxygen saturation levels and cardiopulmonary collapse.

lines 162-171. Having a table/figure showing the MACOCHA score would help make the paper stronger and clearer.  

Thank you for the suggestion. We have added Table 2 which describes the MACOCHA score

I think a paragraph on associated costs (financial; personnel, etc). would make the the paper even stronger and give more food for thought.

Thank you for the suggestion. We have added the following text describing financial considerations in establishing an ART:

Anticipation of the costs of implementation and maintenance of an ART is essential for the program to remain financially viable. Key financial variables to consider include: a) costs of airway equipment and medications; b) the estimated volume of airway consults and procedures; c) personnel costs and d) reimbursement associated with team practice. Equipment costs to account for include (protective personal equipment; i.e., surgical masks, face shield, gown, gloves), disposable airway equipment (e.g. laryngoscope blades and endotracheal tubes, suction catheters, oral airways, SGAs and bag-mask assemblies), medications (i.e., sedatives, neuromuscular blocking agents, and intravenous vasoactive medications), as well as capital expenditures for new equipment such as airway carts with equipment for advanced airway management (i.e. fiberoptic bronchoscopes and video-laryngoscopes). Personnel costs will be based on ART composition as well as the extent to which team members may be engaged in other clinical roles. The volume of emergency airway calls prior to ART implementation will provide information on whether ART members should be solely dedicated to airway responsibilities or may be given additional assignments. ARTs may also serve as consultants for performing other procedures (e.g., obtaining arterial and central venous access) and reimbursement for these services will impact cost considerations. Ongoing education costs, communication equipment or administrative support for data collection and quality improvement should also be considered. The average reimbursement for an emergency airway management consultation multiplied by the frequency of emergency consults should provide a reasonable estimate of total ART reimbursement. Such financial analysis will require periodic reassessment after ART implementation 

Reviewer 3 Report

Thank you for the opportunity to review the review paper regarding the airway response teams for emergency airway management (EAM) in the hospital setting. The authors reviewed the challenges related to EAM and described strategies to improve patients care by implementing an airway response team (ART). Based on latest literatures, they summarized those into eleven theme: the risks associated with EAM, challenges in EAM, rationale for an ART, the team model, activation, identification of high-risk patients, equipment availability, teamwork, implementation, education, and monitoring and quality improvement. This well-written paper is a remarkably interesting one. This review paper provides us recent knowledge about EAM. I would appreciate the authors efforts. I have only two minor issues as below.

Minor issue:

1.     Figure 5

      More graphical figure would be preferable.

2.     Line 316

The reference no. 76 may be no.9.

Author Response

Reviewer 3:

Thank you for the opportunity to review the review paper regarding the airway response teams for emergency airway management (EAM) in the hospital setting. The authors reviewed the challenges related to EAM and described strategies to improve patients care by implementing an airway response team (ART). Based on latest literatures, they summarized those into eleven theme: the risks associated with EAM, challenges in EAM, rationale for an ART, the team model, activation, identification of high-risk patients, equipment availability, teamwork, implementation, education, and monitoring and quality improvement. This well-written paper is a remarkably interesting one. This review paper provides us recent knowledge about EAM. I would appreciate the authors efforts. I have only two minor issues as below.

Minor issue:

  1. Figure 5

      More graphical figure would be preferable.

Thank you. We have revised the figure

  1. Line 316

The reference no. 76 may be no.9.

Thank you. You are correct. We have changed the reference accordingly